# Chemical Composition, In Vitro Bioaccessibility and Antioxidant Activity of Polyphenolic Compounds from Nutraceutical Fennel Waste Extract

**DOI:** 10.3390/molecules26071968

**Published:** 2021-03-31

**Authors:** Luigi Castaldo, Luana Izzo, Stefania De Pascale, Alfonso Narváez, Yelko Rodriguez-Carrasco, Alberto Ritieni

**Affiliations:** 1Department of Pharmacy, University of Naples “Federico II”, 49 Domenico Montesano Street, 80131 Naples, Italy; alfonso.narvaezsimon@unina.it; 2Department of Agricultural Sciences, University of Naples Federico II, 80055 Portici, Italy; depascal@unina.it; 3Laboratory of Food Chemistry and Toxicology, Faculty of Pharmacy, University of Valencia, 46100 Valéncia, Spain; yelko.rodriguez@uv.es; 4Staff of UNESCO Chair on Health Education and Sustainable Development, Federico II University, 80131 Naples, Italy; alberto.ritieni@unina.it

**Keywords:** food waste valorization, health-promoting compounds, polyphenols, bioactive compounds, nutraceutical

## Abstract

Fennel (*Foeniculum vulgare* Mill.) waste contains a broad range of bioactive molecules, including polyphenols, which have poor bioaccessibility during gastrointestinal digestion. This work aimed to investigate the bioaccessibility of total phenolic compounds and the antioxidant capacity during simulated gastrointestinal digestion using two nutraceutical formulations based on non-acid-resistant (NAR) and acid-resistant (AR) capsules containing aqueous-based extracts from fennel waste. Moreover, to obtain a comprehensive investigation of the polyphenolic constituents of the fennel waste extract, a high-resolution mass spectrometry (Q-Orbitrap) analysis was performed. Notably, chlorogenic acids, such as 4-caffeoylquinic acid and 3,4-dicaffeoylquinic acid, were the most detected compounds found in assayed samples (1.949 and 0.490 mg/g, respectively). After in vitro gastrointestinal digestion, the extract contained in AR capsules displayed higher bioaccessibility in both the duodenal and colonic stages (1.96 and 5.19 mg GAE/g, respectively) than NAR capsules (1.72 and 3.50 mg GAE/g, respectively), suggesting that the acidic gastric conditions negatively affected the polyphenol compounds released from the NAR capsules. Therefore, the aqueous extract of fennel waste could be proposed as an innovative and easily available source of dietary polyphenols. Furthermore, the use of an AR capsule could improve the polyphenol bioaccessibility and can be proposed as a nutraceutical formulation.

## 1. Introduction

Fennel (*Foeniculum vulgare* Mill.) is a plant belonging to the Apiaceae family, widely naturalized in many parts of the world and commonly considered as native in the Mediterranean basin [1]. Fennel has been long used as an ingredient in foods, flavor foods, liqueurs, and in the cosmetic industry due to its typical anise-like flavor [2]. The yearly production of fennel has steadily grown over the world, and Italy represents one of the largest producers, manufacturing about 532,000 tons/year [3]. Consequently, colossal amounts of fennel residues are generated in the fennel industry. The overall strategy for the utilization of food wastes is generally built around the production of feeds, fertilizers, and fuels [4]. However, vegetable wastes and byproducts contain a considerable amount of interesting compounds, such as antioxidants, which may be recovered and used effectively as ingredients in the design of new healthy foods or in nutraceutical formulations [5]. Typically, the use of a simple water-based extraction procedure is recognized as the greenest way to recover bioactive compounds and reduce the environmental impact of agro-industrial wastes [6,7]. The outer part of the fennel represents the main residue in the fennel industry. Previous scientific investigations suggest that fennel waste contains a wide range of human health-related compounds, including polyphenols [8]. Dietary polyphenols are secondary metabolites of plants naturally occurring in fruits, vegetables, tea, coffee, among others [9]. Polyphenols are well-known to exert a broad range of important biological properties able to reduce oxidative stress and inflammation [10]. In this line, a wealth of data from epidemiological studies support an inverse association between polyphenol-rich diets and the incidence of several age-related diseases [11]. Despite the efforts made in the last decade to find an alternative use for food waste and by-products, fennel waste remains unused as a raw material to produce high value-added products. The most studied compounds of fennel waste are extracted from the volatile fraction, whereas the polyphenolic compounds have been barely evaluated to date [12]. Recently, Di Donato et al. [8] investigated the polyphenolic profile of fennel waste through reverse-phase high-performance liquid chromatography (RP-HPLC) and LC-mass spectrometry (MS) technologies. The results showed that flavonoids and some important hydroxycinnamic acids were the most common polyphenols in fennel waste. Scientific evidence suggests that the above-mentioned active compounds, in addition to their well-established antioxidant capacity, exert a wide range of other important biological properties, such as anti-inflammatory and anticancer activities, as well as the ability to modulate cell signaling and gene expression in several experimental models [13], becoming good candidates for nutraceutical applications [14].

To fully exert their biological properties, polyphenols need to be available for absorption in the target tissue [15]. Moreover, several studies have shown that the absorption of polyphenols occurs mainly in the colon stage after metabolized by gut microbiota [16]. As reported, polyphenols are highly sensitive to the action of digestive enzymes, bacterial microflora, temperature, and pH during gastrointestinal digestion that could affect their bioaccessibility as well as a beneficial effect on health [17]. In order to overcome this limitation, potential valid strategies have been proposed [18]. A way to improve the polyphenols bioaccessibility may be the use of nutraceutical forms, protecting chemical characteristics of polyphenols that could be affected during gastrointestinal digestion, as lately evidenced by literature [18].

Among the analytical methods, the most widely employed technique for the polyphenol detection in vegetal matrices is based on LC separation combined with MS and tandem MS (MS/MS) [19]. Notwithstanding that, the high-resolution mass spectrometers (HRMS), such as Q-Orbitrap, coupled with ultra-HPLC (UHPLC), represent an optimal choice for the identification and quantification of a, which range of bioactive compounds in foodstuffs compounds in plant matrices due to its high-resolution power, high peak efficiency, short analysis time, and quantification based on accurate mass measurement [20,21].

Hence, this research aimed to provide a comprehensive analysis of the polyphenolic profile of the aqueous extract of fennel waste through UHPLC-Q-Orbitrap HRMS analysis. The bioaccessibility of total phenolic compounds (TPC) and the antioxidant capacity during an in vitro gastrointestinal digestion were also assessed using two nutraceutical formulations based on non-acid-resistant (NAR) and acid-resistant (AR) capsules containing aqueous extracts of fennel waste in order to provide useful data on this innovative source of active molecules and their efficacy in different nutraceutical formulations.

## 2. Results

### 2.1. Not-Target Identification of Flavonoids and Phenolic Compounds in the Aqueous Extract of Fennel Waste through UHPLC-Q-Orbitrap HRMS

UHPLC-Q-Orbitrap HRMS analysis was performed to identify flavonoids and phenolic acids in the aqueous extract of fennel waste. There were identified up to 24 phenolic compounds in the assayed samples. Optimal separation of the investigated compounds was achieved in a total run time of 13 min. Mass parameters, including retention time (RT), ion assignment, theoretical mass (*m/z*), measured mass (*m/z*), product ion, accuracy and sensitivity, are reported in Table 1. The structural isomers identification of the epigallocatechin and gallocatechin (*m/z* 305.06675); epicatechin and catechin (*m/z* 289.07175); myricitrin and isoquercetin (*m/z* 463.08820) was carried out by comparing their RT and fragmentation pattern with data reported in published articles.; All experiments were performed in electrospray ionization negative (ESI^−^) mode. The obtained relative intensity of product ion was from 80–100%, whereas the parent compound was held at 10% of intensity. All sample data were captured in full-scan mass spectra, allowing the untargeted identification of compounds and retrospective data analysis.

### 2.2. Quantification of Flavonoids and Phenolic Compounds in the Aqueous Extract of Fennel Waste

The identified polyphenols (*n* = 24) found in the aqueous extract of fennel waste were characterized using a UHPLC coupled to a high-resolution Orbitrap mass spectrometry. Calibration curves were built for the quantitative analysis of the found analytes. Regression coefficients >0.990 for all the analytes were obtained. Some important flavonoids (*n* = 18), including isoflavones, flavonols, flavanones, flavanols, and flavones, were found in the analyzed extracts from fennel waste. Among flavonoids, ellagic acid was the compound found at the highest concentration in the extracts, with a mean value of 0.101 mg/g (Table 2). In the aqueous extract of fennel waste here investigated, catechin represented 75% of total flavanols detected, with a mean value of 0.021 mg/g. Regarding the flavones group, luteolin was the predominant compound identified, with a mean content of 0.024 mg/g. As far as flavanones are concerned, naringenin was the most abundant analyte quantified in the here-investigated extracts at a mean value of 0.018 mg/g. On the other hand, the main phenolic acids, including benzoic acid (syringic and protocatechuic acids) and cinnamic acids (caffeic, 4-caffeoylquinic (4-CQA), 3,4-dicaffeoylquinic (3,4-diCQA), ferulic and, *p*-coumaric acids) were evaluated in the extracts. Among all polyphenols investigated, the analytes belonging to the cinnamic acids group were the most abundant compounds found in the assayed extracts, with a sum concentration of 2.745 mg/g. Moreover, 4-CQA was detected at a level significantly higher than the other assayed cinnamic acids, with a mean value of 1.949 mg/g. Concerning benzoic acids, both syringic and protocatechuic acids were detected in concentration <LOQ, as shown in Table 2.

### 2.3. In Vitro Bioaccessibility of Aqueous Extract of Fennel Waste in NAR and AR Capsules

An in vitro gastrointestinal digestion performed on NAR and AR capsules containing 500 mg of aqueous extract of fennel waste was done in order to provide useful data on their ability to protect polyphenols during gastrointestinal digestion. Therefore, fennel extract polyphenols bioaccessibility on NAR and AR capsules was assessed and compared with the nondigested extract. Total phenolic compounds were detected through the Folin–Ciocâlteu method in each phase of the in vitro gastrointestinal digestion. As shown in Table 3, the oral bioaccessibility obtained from both NAR and AR capsules was not relevant, whereas, in the gastric phase, it was not relevant only for the AR sample. The extract in AR capsules showed a significantly higher TPC value (*p*-value ≤ 0.05) in the duodenal phase when compared with NAR samples (1.96 and 1.72 mg GAE/g, respectively). Moreover, a similar trend was also observed in the colonic phase (considered as Viscozyme L phase plus Pronase E phase), where the extract in AR capsules displayed a significantly higher TPC (*p*-value ≤ 0.05) value when compared with NAR ones (5.19 and 3.50 mg GAE/g, respectively).

### 2.4. Antioxidant Capacity of the Aqueous Extract of Fennel Waste Contained on NAR and AR Capsules after Simulated Gastrointestinal Digestion

The changes in antioxidant activity release during the gastrointestinal tract of the aqueous extract of fennel waste contained on NAR and AR capsules were assessed using three different assays (DPPH, ABTS, and FRAP) in order to choose the best delivery system able to protect the antioxidant compounds until reaching the target tissues. Table 4 displays the data as millimoles of Trolox equivalent per kilogram of extract (average value and SD) found in each one of in vitro digestion phase.

The results highlighted that in all assayed methods, the digested samples showed a lower antioxidant capacity compared to nondigested extract (*p*-value ≤ 0.05). On the other hand, the highest antioxidant activity along the simulated gastrointestinal digestion was displayed from both NAR and AR samples in the colonic phase. As shown in Table 4, the highest antioxidant activity observed in this stage was found from AR samples in all spectrophotometric tests.

Moreover, strong positive correlations were found among antioxidant activity evaluated by the DPPH, ABTS, and FRAP values and their corresponding data obtained from the TPC test evaluated after gastrointestinal digestion, as shown in Appendix A.

## 3. Discussion

This study aimed to provide useful data about the active molecules present in the fennel waste, such as polyphenols, in order to add value to this agro-industry waste material. A simple aqueous-based extraction was used to obtain a food-grade extract from fennel waste. Overall, the results obtained in the current study indicate that the fennel waste may represent an innovative source of active compounds, including some important phenolic compounds belonging to the cinnamic and benzoic acids, such as 4-CQA 3,4-diCQA, caffeic acid, ferulic acid, protocatechuic acid, and gallic acid. Moreover, the main flavonoids, including isoflavones, flavonols, flavanones, flavanols, and flavones, were found in the water-based extracts from fennel waste. A large amount of scientific data have reported that habitual dietary consumption of these molecules may play a key role in the prevention of several age-related human diseases, including cardiovascular pathologies, type-2 diabetes, cancer, and obesity as well [22,23].

Nowadays, the valorization of food waste and byproducts from agriculture and other food sectors is growing rapidly due to the well-known opportunities to recover a wide range of high added-value compounds [24]. Data obtained from several in vitro tests highlighted that various food waste and by-products, including olive mill wastewater, olive leaves, winemaking waste, brewing spent grains, apple pomace, among others, could be used as a sustainable source of bioactive molecules for cosmeceutical and nutraceutical applications [25].

In spite of several investigations that have been carried out on the phenolic profile of fennel, the study of fennel waste is limited to the volatile fraction, whereas the polyphenolic constituents have been barely evaluated in this waste material making it difficult to compare results. To the best of our knowledge, this is the first work reporting a comprehensive polyphenolic profile of aqueous-based extract of fennel waste achieved by using the HRMS Q-Orbitrap methodology. In previous works, Parejo et al. [26] investigated the polyphenol fraction of the fennel residue obtained after the distillation of the essential oil. The authors identified eight phenolic compounds and concluded that fennel waste might be used as a novel source of antioxidants. Similarly, Pacifico et al. [27] evaluated the phenolic profile of different extracts of fennel leaves, reporting appreciable amounts of quercetin-O-glycoside and 3-CQA. Total polyphenol content found in assayed extracts was similar to those reported by Di Donato et al. [8] for an ethanol-based extract of fennel waste obtained with ultrasound-assisted extraction (UAE) procedure, which showed a total polyphenol content up to 4.79 mg/g. The quantified compounds were five of twenty-four compounds (*p*-coumaric, ferulic, caffeic, chlorogenic acids, and quercetin) monitored in the assayed samples.

Although scientific evidence has reported that polyphenols are responsible for several health-promoting effects [28], it is crucial to consider that these important plant-based secondary metabolites are highly sensitive, and gastrointestinal digestion can affect the biological properties and structure of these compounds, limiting their bioaccessibility [29]. According to Rocchetti et al. [30], bioaccessibility refers to the amount of water-soluble phytochemicals release in each one of the digestion stages from the complex food material, which could be considered absorbable by the human body.

Considering the polyphenols susceptibility to digestion conditions, the use of NAR and AR capsules in the formulation of dietary supplement products may represent an effective strategy to move active molecules to the target tissues; in this way, the bioactive compounds can be absorbed and fully exert their biological properties [31]. In this sense, Tenore et al. [32] reported that about 50% of native polyphenols present in tea samples were lost after the gastric phase and 92% during the intestinal digestion.

Therefore, the bioaccessibility of the total phenolic compounds and the variation of antioxidant capacity was also assessed using an in vitro digestion protocol performed on two nutraceutical formulations based on NAR and AR capsules containing aqueous-based extract from fennel waste. The protocol used to simulate human gastrointestinal digestion in the present study was recently developed in the COST action INFOGEST network [33]. The above protocol is recognized as the most suitable method for comparing results across research teams. Since the fermentation of the large intestine is not covered in the INFOGEST protocol, the combined action of Viscozyme L and Pronase E has been used to simulate the colon digestion process [34]. Viscozyme L contains a mix of several carbohydrases, such as xylanase, β-glucanase, hemicellulose, glucanase, arabinose, and cellulase, whereas Pronase E is a mixture of bacterial protease. Previous studies have proposed the use of Pronase E and Viscozyme L to reproduce the activity of the gut microbiota as an effective alternative to the use of fecal inoculum [35]. On the other hand, several strategies have been reported to evaluate the polyphenol bioaccessibility after simulated gastrointestinal digestion. The use of immortalized cell lines, such as Caco-2, represents the most common technique for assessing polyphenol bioaccessibility [36]. However, in order to obtain an overview of polyphenol bioaccessibility, in the present work, TPC by the Folin–Ciocâlteu assay was measured after all phases of gastrointestinal digestion. The results highlighted strong correlations among the data obtained from DPPH, ABTS, and FRAP assays and TPC values assessed during in vitro gastrointestinal digestion, supporting that the assayed tests provide some reliable information on the antioxidant compounds released by the nutraceutical formulations assayed during the simulated gastrointestinal process.

The highest percentage of decrease in TPC value and antioxidant activity from not-digested samples was observed during the gastric phase in NAR samples, suggesting that the acidic gastric conditions affected the polyphenol compounds present in the extracts. On the other hand, the lowest percentage of decrease in both TPC value and antioxidant activity was found in AR samples after the colonic stage. The AR capsules in HPMC have been recognized as highly efficient in protecting components during the gastric passage, while gelatin capsules are not resistant to acidic gastric conditions [37]. As expected, the oral and gastric bioaccessibility reported from AR capsules was 0 mg GAE/g, suggesting that the capsules were not affected during these digestion stages. However, for the NAR samples, only the oral bioaccessibility was 0 mg GAE/g, whereas, in the gastric phase, there was a release of substances contained in them. Interestingly, the extract contained in AR capsules showed a significantly higher TPC value and antioxidant activity in both the duodenal and colonic phases (considered as Viscozyme L phase plus Pronase E phase) when compared with NAR samples.

Interestingly, Izzo et al. [38] have reported that red cabbage extract encapsulated in AR capsule after in vitro gastrointestinal digestion showed higher colon bioaccessibility when compared to the extract digested without a capsule. Similar results have been observed by Amrani-Allalou et al. [39], who reported a strong decrease in TPC value during gastrointestinal digestion in non-encapsulated medicinal plant extracts compared to the same extracts containing in AR capsules. These outcomes show that the AR capsules are able to protect the polyphenol fraction from the gastric condition, protecting the chemical characteristics of the bioactive molecules. Therefore, AR capsules can represent a valid strategy to move active molecules to the target tissues for exerting their nutraceutical potential.

## 4. Materials and Methods

### 4.1. Reagents and Materials

Waste solids residues of *F. vulgare* Mill. (Apiaceae) remaining after the recovery of the bulb from plants was provided from F.lli Napolitano C. e G. (Società Agricola Snc) Salerno, Italy. All samples were grown in the Campania region, South of Italy, and harvested in December 2020. The fennel wastes (three many ten kilograms each) consisted of foliar parts and white outer sheaths. The outer parts of the fennel with visible spoilage were removed. All materials were freeze-dried and homogenized through a laboratory mill and then stored at −80 °C until further analysis.

The standards of polyphenols (purity > 98%), namely 3,4-diCQA, 4-CQA, protocatechuic acid, caffeic acid, syringic acid, *p*-coumaric acid, ferulic acid, ellagic acid, epigallocatechin, epicatechin, gallocatechin, catechin, vitexin, genistein, isoquercetin, rutin, myricitrin, diosmin, myricetin, quercetin, naringenin, and luteolin were acquired from Sigma-Aldrich (Milan, Italy).

Potassium chloride (KCl), 2′2-azino-bis-3-ethylbenzthiazoline-6-sulfonic acid, TPTZ, (±)-6-hydroxy-2,5,7,8-tetramethylchromane-2-carboxylic acid, 1,1-diphenyl-2-picrylhydrazyl, sodium acetate (NaCH_3_COO), calcium chloride dihydrate (CaCl_2_ · 2H_2_O), sodium hydroxide (NaOH), monosodium phosphate (NaH_2_PO_4_) potassium thiocyanate (KCNS), sodium sulfate (NaSO_4_), sodium bicarbonate (NaHCO_3_), sodium chloride (NaCl), and potassium persulfate (K_2_S_2_O_8_) were provided from Sigma-Aldrich (Milan, Italy) and also enzymes pancreatin (8 × USP) from porcine pancreas, α-amylase (1000–3000 U/mg solid) from human saliva, bacterial protease from *Streptomyces griseus* commonly called Pronase E (≥3.5 U/mg solid), pepsin (≥2500 U/mg solid) from porcine gastric mucosa, and Viscozyme L were purchased from Sigma-Aldrich (Milan, Italy).

Methanol (MeOH) water (UHPLC-MS grade) and formic acid (FA) were purchased from Merk (Darmstadt, Germany), whereas deionized water (<18 MX cm resistivity) was provided from a Milli-Q water purification system (Millipore, Bedford, MA, USA).

### 4.2. Polyphenols Extraction

Aqueous extract of polyphenolic compounds from fennel waste was obtained following the procedure described by Pacifico et al. [27] with slight adaptations. Briefly, 10 g of freeze-dried samples were suspended in 200 mL of deionized water. The mixture was placed in a shaking water bath (1086, GFL, Rome, Italy) for 30 min at 80 °C and stirred at 100× *g*. Afterward, the mixture was sonicated for 30 min and then centrifuged at 4900× *g* for 10 min. The supernatant was collected, and the pellets were re-extracted following the same procedure. After this, the pooled supernatants were lyophilized, and the obtained powder was employed for the formulation of the capsules. The capsules used were AR (pharmaceutical-grade hydroxypropyl methylcellulose) and NAR (pharmaceutical-grade gelatin capsules), containing 500 mg of fennel waste polyphenolic extract.

### 4.3. Ultra-High-Performance Liquid Chromatography and Orbitrap HRMS Analysis

Chromatographic separation was carried out through a UHPLC (Dionex UltiMate 3000, Thermo Fisher Scientific, Waltham, MA, USA), prepared with a degassing system, an autosampler device, and a quaternary UHPLC pump (maximum pressure tolerance = 1250 bar) [40].

Separation of analytes occurred with a thermostated (T = 25 °C) Kinetex F5 (50 × 2.1 mm, 1.7 µm particle size, Phenomenex, Torrance, CA, USA) column. The mobile phases were water (A) and MeOH (B). Both phases were prepared at 0.1% of FA. The injection volume was 5 µL. The gradient elution started with 0% B for 1 min, increased to 80% B in 2 min, and then increased again, reaching 100% B in 3 min. Afterward, the gradient switched back to 0% B in 2 min and maintained for 2 min for column re-equilibration. The flow rate was 0.5 mL/min. Mass spectrometry was carried out through a Q-Exactive Orbitrap mass spectrometer (Thermo Fisher Scientific, Waltham, MA, USA) combined with an ESI source operating in negative ion mode. The acquisition was conducted by setting two scan events: all ion fragmentation (AIF) and full ion MS. AIF mode conditions were: mass resolving power to 17,500 full widths at half-maximum (FWHM), maximum injection time to 200 ms, automatic gain control (AGC) target 1 × 10^6^, scan time 0.10 s, isolation window to 5 *m/z*, scan range 80–1200 *m/z*, and retention time to 30 s. The collision energies were included in the range between 10 and 60 eV. The full MS mode was performed considering the following conditions: resolution power of 70,000 FWHM, scan range 80–1200 *m/z*, AGC target 1 × 10^6^, injection time set to 200 ms, and scan rate at 2 scan/s. The ion source conditions were: capillary temperature 320 °C and spray voltage 3.5 kV.

Quantification was performed considering the exact mass with a mass error of 5 ppm in both AIF mode and full scan MS. Data treatment and processing were performed through Xcalibur software 3.1.66.19 (Xcalibur, Thermo Fisher Scientific, Waltham, MA, USA).

### 4.4. In Vitro Gastrointestinal Digestion

Nutraceutical formulations of aqueous-based extracts from fennel waste in AR and NAR capsules were subjected to in vitro gastrointestinal digestion following the procedure developed by Minekus et al. [33] in order to evaluate the change in polyphenol bioaccessibility and antioxidant activity during the different stages of gastrointestinal digestion. The simulated salivary (SSF), gastric (SGF), and intestinal fluids (SIF) were prepared according to the previously described procedure by Minekus et al. [33] and reported in Appendix A.

In short, aqueous-based extract from fennel waste in AR and NAR capsules was suspended in 3.5 mL of warmed SSF, 0.5 mL of α-amylase solution, 975 µL of water and 25 µL of 0.3 M CaCl_2_ (H_2_O)_2_. Afterward, NaOH 1 M was added to the mixture in order to adjust the pH to 7 before being incubated for 2 min at 37 °C at 100× *g* in a shaker bath.

Then, in order to simulate the gastric condition, 7.5 mL of SGF, 5 µL of 0.3 M CaCl_2_ (H_2_O)_2_, 695 µL of water, and 1.6 mL pepsin solution were added and thoroughly mixed. The solution was incubated for 120 min at 37 °C in a shaker bath at 100× *g*. After this, the mixture was adjusted at pH 3 with HCl 1 M.

Then, in order to mimic the intestinal phase, 5 mL pancreatin solution (100 U/mL of trypsin activity), 11 mL of SIF, 40 µL of 0.3 M CaCl_2_ (H_2_O)_2_, 1300 µL of H_2_O, and 2.5 mL bile salt solution (65 mg/mL) were added and mixed. The pH of the mixture was increased to 7 with a solution of 1 M NaOH. The sample was incubated for 2 h at 37 °C in an orbital shaker bath at 100× *g* and then centrifuged at 4900× *g* for 10 min at 37 °C.

Finally, the samples were treated following the procedure described elsewhere [34] to simulate the colon stage. Briefly, 5 mL of 1 mg/mL Pronase E solution were added to the remaining pellets. The mixture was incubated (pH 8, for 60 min) in a shaker bath at 37 °C at 100× *g*. Then, 150 µL of Viscozyme L and 5 mL of water were added to the solution and incubated for 16 h (pH 4) in a shaker bath at 37 °C at 100× *g*. 500 µL of the supernatants were collected, freeze-dried and stored at −18 °C until analysis.

### 4.5. Determination of Antioxidant Capacity

The antioxidant capacity of the fennel waste extract subjected to the gastrointestinal digestion in the AR and NAR capsules was determined and compared through three different spectrophotometric assays. The results were expressed as mmol of Trolox per kg of sample.

#### 4.5.1. ABTS Assay

The ABTS radical-scavenging assay was measured by using the methodology reported by Dini et al. [41], with minor modifications. In short, 2.5 mL of ABTS (7 mM) were added to 44 µL of potassium persulfate (2.5 mM). The mixture was thoroughly mixed and then kept for 16 h at room temperature. Then, the mixture was diluted with ethanol to reach 734 nm an absorbance value of 0.70 (±0.02). One-hundred µL of the opportunely diluted samples were added to 1000 µL of ABTS radical working solution. The decrease in absorbance was measured after 3 min at 734 nm.

#### 4.5.2. DPPH Assay

The antioxidant activity of the analyzed samples was carried out using the DPPH procedure suggested by Dini et al. [42] with minor modifications. In brief, 1 mg of DPPH was diluted in MeOH until the absorbance value reached 0.90 (±0.02) at 517 nm. Then, 200 µL of the opportunely diluted samples were added to 1 mL of DPPH radical working solution and mixed. The absorbance was monitored after 10 min at 517 nm.

#### 4.5.3. FRAP Assay

The FRAP assay was carried out following the procedure described in a previous work et al. [43] with some modifications. Briefly, the FRAP solution was prepared by mixing 1.25 mL of ferric chloride solution (20 mM, H_2_O), 1.25 mL of 2,3,5-triphenyltetrazolio chloride (TPTZ) solution (10 mM) in HCl (40 mM), and 12.5 mL of acetate buffer (0.3 M, pH 3.6). Then, 2.85 mL of FRAP reagent was added to 150 µL of samples properly diluted and mixed. The absorbance was measured after 4 min at 593 nm.

### 4.6. Total Phenolic Content

Determination of TPC was performed according to the procedure performed by Izzo et al. [44] with slight modifications. In short, 125 µL of the sample were added to 500 µL of deionized water, and 125 µL of the Folin–Ciocâlteu reagent 2 N. After incubation for 6 min at room temperature, 1.25 mL of a 7.5% sodium carbonate solution, and 1 mL of deionized water were added to the mixture and mixed. The absorbance at 760 nm after 90 min was recorded. Results were expressed as mg of GAE per gram of sample.

### 4.7. Statistical Analysis

Statistical analysis of the data was performed through Stata 12 software (StataCorp LP, College Station, TX, USA). The differences between mean values were evaluated by using Tukey’s test at the level of significance *p*-value ≤ 0.05. All experiments were performed in triplicate, and the results were displayed as average ± SD. The correlation coefficients were evaluated by using Pearson’s method.

## 5. Conclusions

In summary, this study provided an in-depth analysis of the polyphenolic fraction, including phenolic acids (*n* = 7) and flavonoids (*n* = 17) in water-based extracts from fennel waste using a UHPLC Q-Orbitrap high-resolution mass spectrometry. Data obtained highlighted that cinnamic acids, such as 4-CQA and 3,4-diCQA, were the predominant phenolic compounds detected in the assayed extracts. Moreover, our results suggested that during simulated gastrointestinal digestion, AR capsules were able to protect bioactive compounds from the negative effect of gastric conditions, resulting in a significantly higher TPC value and antioxidant activity in both the duodenal and colonic phases than NAR capsules. Data highlighted that the aqueous extract from fennel waste could be considered as an innovative source of dietary polyphenols, and the extract encapsulated in AR capsules could be an effective strategy to move the antioxidants to the target tissues.

Nevertheless, further studies useful to identify the several metabolites generated during the gastrointestinal process are needed to better understand the possible benefits on human health.

## Figures and Tables

**Table 1 molecules-26-01968-t001:** Chromatographic and spectrometric parameters of the investigated compounds (*n* = 24).

Compound	RT (min)	Adduct Ion	Chemical Formula	Theoretical Mass *(m/z*)	Measured Mass (*m/z*)	Product Ion	Accuracy (Δ ppm)	LOD (mg/kg)	LOQ (mg/kg)
Protocatechuic acid	2.31	[M-H]^−^	C_7_H_6_O_4_	153.01930	153.01857	109.0284	−4.77	0.013	0.039
Epigallocatechin	2.84	[M-H]^−^	C_15_H_14_O_7_	305.06675	305.0665	219.06580; 159.10190; 121.02846; 109.02807	−0.82	0.026	0.078
4-Caffeoylquinic acid	3.00	[M-H]^−^	C_16_H_18_O_9_	353.08780	353.08798	191.05594; 84.98998	0.51	0.013	0.039
Epicatechin	3.17	[M-H]^−^	C_15_H_14_O_7_	289.07176	289.07202	221.94647; 203.09201; 161.04478	0.90	0.013	0.039
Gallocatechin	3.19	[M-H]^−^	C_15_H_14_O_8_	305.06676	305.06681	219.06254; 159.10185; 109.02836; 121.02847	0.16	0.013	0.039
Caffeic acid	3.23	[M-H]^−^	C_9_H_8_O_4_	179.03498	179.03455	134.99960	−2.40	0.013	0.039
Catechin	3.34	[M-H]^−^	C_15_H_14_O_6_	289.07175	289.07205	247.02241; 205.10712; 151.03923; 125.02335	1.04	0.026	0.078
Syringic acid	3.36	[M-H]^−^	C_9_H_10_O_5_	197.04555	197.04503	182.02153; 166.99791	−2.64	0.026	0.078
*p*-cumaric acid	3.46	[M-H]^−^	C_9_H_8_O_3_	163.04001	163.03937	119.04917	−3.92	0.013	0.039
Vitexin	3.48	[M-H]^−^	C_21_H_20_O_10_	431.09837	431.09711	341.10803; 311.05457; 269.13815	−2.92	0.013	0.039
3,4-Dicaffeoylquinic acid	3.51	[M-H]^−^	C_16_H_18_O_12_	515.11950	515.11993	353.08667; 191.94507	0.83	0.026	0.078
Ferulic acid	3.55	[M-H]^−^	C_10_H_10_O_4_	193.05063	193.05016	178.02666; 149.06009; 134.99963	−2.43	0.026	0.078
Naringin	3.56	[M-H]^−^	C_27_H_32_O_14_	579.17193	579.17212	459.09421; 339.03604; 271.04913	0.33	0.013	0.039
Rutin	3.59	[M-H]^−^	C_27_H_30_O_16_	609.14611	609.14673	300.99911; 271.05026; 255.12390	1.02	0.013	0.039
Isoquercetin	3.61	[M-H]^−^	C_21_H_20_O_12_	463.08820	463.08853	431.09848; 187.09698; 174.95542	0.71	0.026	0.078
Myricitrin	3.62	[M-H]^−^	C_21_H_20_O_12_	463.08820	463.08701	316.02126; 178.97646	−2.57	0.013	0.039
Diosmin	3.64	[M-H]^−^	C_28_H_31_O_15_	607.16684	607.16534	300.99796; 284.03838	−2.47	0.013	0.039
Ellagic acid	3.65	[M-H]^−^	C_14_H_6_O_8_	300.99899	300.99911	245.91669; 29.93712; 185.01208; 117.00336	0.40	0.013	0.039
Kaempferol 3 glucoside	3.68	[M-H]^−^	C_21_H_20_O_11_	447.09195	447.09329	284.03079; 255.02881; 227.07033	3.00	0.013	0.039
Myricetin	3.73	[M-H]^−^	C_14_H_10_O_8_	317.03029	317.02924	178.87917; 151.00217; 137.02290	−3.31	0.013	0.039
Quercetin	3.88	[M-H]^−^	C_15_H_10_O_7_	301.03538	301.03508	273.04007; 174.95551	−1.00	0.013	0.039
Naringenin	3.91	[M-H]^−^	C_15_H_12_O_5_	271.0612	271.0611	235.92595; 151.03917	−0.37	0.013	0.039
Luteolin	3.98	[M-H]^−^	C_15_H_10_O_6_	285.04046	285.04086	174.95486; 89.02095	1.40	0.026	0.078
Genistein	4.05	[M-H]^−^	C_15_H_10_O_5_	269.04554	269.04562	241.14435; 213.14908; 151.03935	0.30	0.013	0.039

Abbreviations: LOD: limit of detection; LOQ: limit of quantification.

**Table 2 molecules-26-01968-t002:** Polyphenol content in the assayed extracts. The results are displayed as average value (mg/g) and standard deviation (SD).

Compounds	Average (mg/g)	SD
PHENOLIC ACIDS		
*Cinnamic acid*		
Caffeic acid	0.049	0.002
4-CQA	1.949	0.142
3,4 diCQA	0.490	0.035
Ferulic acid	0.258	0.001
*p*-Coumaric acid	<LOQ	
SUM	2.745	0.067
*Benzoic Acid*		
Syringic acid	<LOQ	
Protocatechuic acid	<LOQ	
FLAVONOIDS		
*Flavones*		
Luteolin	0.024	0.004
Vitexin	<LOQ	
Diosmin	<LOQ	
Kaempferol 3 glucoside	<LOQ	
SUM	0.024	0.004
*Flavanols*		
Catechin	0.021	0.002
Epicatechin	0.007	0.001
Epigallocatechin	<LOQ	
Gallocatechin	<LOQ	
SUM	0.028	0.002
*Flavanones*		
Naringenin	0.018	0.001
Naringin	<LOQ	
SUM	0.018	0.001
*Flavonols*		
Quercetin	0.019	0.005
Isoquercetin	0.009	0.002
Rutin	<LOQ	
SUM	0.028	0.004
*Isoflavone*		
Genistein	<LOQ	
Myricetin	0.021	0.001
Myricitrin	0.039	0.001
SUM	0.060	0.002
*Hydrolyzable tannins*		
Ellagic acid	0.101	0.003
TOTAL POLYPHENOLS	5.824	0.037

Statistical differences were evaluated by Tukey’s test; *p*-value ≤ 0.05 was considered significant (*n* = 3).

**Table 3 molecules-26-01968-t003:** Total phenolic content in the investigated samples.

Samples	TPC mg GAE/g ± SD	
**Not Digested**	8.22 ± 0.31	-
**-**	**NAR**	**AR**
Digestion Stage		
Oral stage	n.d.	n.d.
Gastric stage	1.04 ± 0.12	n.d.
Duodenal stage	1.72 ± 0.11	1.96 ± 0.05
Pronase E	1.86 ± 0.16	3.26 ± 0.27
Viscozyme L	1.64 ± 0.12	1.93 ± 0.06
*Total colonic stage*	3.50 ± 0.14	5.19 ± 0.15

Statistical differences were evaluated by Tukey’s test; *p*-value ≤ 0.05 was considered significant (*n* = 3). n.d.: not detected.

**Table 4 molecules-26-01968-t004:** Antioxidant activity evaluated by DPPH, ABTS and FRAP assays.

-	DPPH mmol/kg ± SD		ABTS mmol/kg ± SD		FRAP mmol/kg ± SD	
**Not Digested**	14.2 ± 0.9	-	17.7 ± 1.4	-	12.3 ± 1.1	-
-	**AR**	**NAR**	**AR**	**NAR**	**AR**	**NAR**
*Digestion stage*						
Oral stage	n.d.	n.d.	n.d.	n.d.	n.d.	n.d.
Gastric stage	n.d.	1.36 ± 0.2	n.d.	1.7 ± 0.2	n.d.	0.7 ± 0.1
Duodenal stage	2.2 ± 0.4	1.9 ± 0.4	3.6 ± 0.5	2.9 ± 0.3	1.4 ± 0.2	1.1 ± 0.1
Pronase E stage	4.2 ± 0.1	2.1 ± 0.2	5.4 ± 0.3	4.0 ± 0.2	4.2 ± 0.4	3.2 ± 0.2
Viscozyme L stage	1.8 ± 0.7	1.6 ± 0.1	3.0 ± 0.4	2.5 ± 0.1	3.5 ± 0.2	2.2 ± 0.1
*Total colonic stage*	5.0 ± 0.4	3.7 ± 0.2	8.4 ± 0.4	6.5 ± 0.2	7.7 ± 0.3	5.4 ± 0.2

Statistical differences were evaluated by Tukey’s test; *p*-value ≤ 0.05 was considered significant (*n* = 3). n.d.: not detected.

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
