# Peer review of "Chemical Composition, In Vitro Bioaccessibility and Antioxidant Activity of Polyphenolic Compounds from Nutraceutical Fennel Waste Extract"

_molecules, 2021, doi:10.3390/molecules26071968_

Round 1

Reviewer 1 Report

This study describes the phenolic compounds of fennel waste extract using appropriate methodologies and the all MS sound very good from a scientific standpoint. Authors examined two nutraceutical formulations based on non-acid resistant (NAR) and acid-resistant (AR) capsules. Overall I recomend accepting the manuscript after a minor revision.

I think, that the goal stated in the abstract „aimed to investigate the bioaccessibility of total phenolic compounds” was not achieved, as only in vitro studies have been performed with the two formulas. In contrast, the wording of the title is precise: in vitro bioaccessibility was studied.

Questions:

  1. Several in vitro digestion models simulating the human gastrointestinal tract have been developed to assess bioaccessibility and the experimental designs of the different digestion systems are distinct. Could you explain, why this system was chosen for present study?
  2. What was the standard deviation of the MS measurements? Is it justified to enter 5 decimal places for the measured m / z data (Table 1)?
  3.  “in vitro” should be italic, check throughout the text.

Author Response

Response to reviewer 1

Manuscript ID: molecules-1157736

Type of manuscript: Article
Title: Chemical Composition, in Vitro Bioaccessibility and Antioxidant Activity of Polyphenolic Compounds from Nutraceutical Fennel Waste Extract

This study describes the phenolic compounds of fennel waste extract using appropriate methodologies and the all MS sound very good from a scientific standpoint. Authors examined two nutraceutical formulations based on non-acid resistant (NAR) and acid-resistant (AR) capsules. Overall I recomend accepting the manuscript after a minor revision.

I think, that the goal stated in the abstract „aimed to investigate the bioaccessibility of total phenolic compounds” was not achieved, as only in vitro studies have been performed with the two formulas. In contrast, the wording of the title is precise: in vitro bioaccessibility was studied.

Questions:

Point 1: Several in vitro digestion models simulating the human gastrointestinal tract have been developed to assess bioaccessibility and the experimental designs of the different digestion systems are distinct. Could you explain, why this system was chosen for present study?

Response 1:  - As suggested by reviewer 1, the authors added the missing information in the manuscript. “The protocol used in the present study to simulate human gastrointestinal digestion was recently developed in the COST action INFOGEST network [33]. The above protocol is recognized as the most suitable method for comparing results across research teams. Since the fermentation of the large intestine is not covered in the INFOGEST protocol, the combined action of Viscozyme L and Pronase E has been used to simulate the colon digestion process [34]. Viscozyme L contains a mix of several carbohydrases such as xylanase, β-glucanase, hemicellulose, glucanase, arabinose, and cellulase, whereas Pronase E is a mixture of bacterial protease. Previous studies have proposed the use of Pronase E and Viscozyme L to reproduce the activity of the gut microbiota as an effective alternative to use of fecal inoculum [35].”

Point 2: What was the standard deviation of the MS measurements? Is it justified to enter 5 decimal places for the measured m / z data (Table 1)?

Response 2:  As suggested by reviewer 1, the authors reduced the decimal places for the measured m/z data (Table 1) from 5 to 2.

Point 3: “in vitro” should be italic, check throughout the text.

Response 3:  - As suggested by reviewer 1, the authors changed the word “in vitro” to “in vitro”.

The authors thank reviewer 1 for evaluating our manuscript.

Reviewer 2 Report

  • Table 1: provide the full name for LOQ and LOD abbreviations, as it is not mentioned anywhere in text.
  • lines340-341: correct the phrase"a previously work et al., [35] with some modifications". 
  • for AR and NAR capsules suitable reference should be incorporated in text, preferably within paragraph 4.4, justifying the acid and non acid-resilient character of each capsule.
  • minor corrections: line 28 correct " acid-gastric condition" to acidic gastric cinditions. Line 348 change 0.125 milliliters to 125 μl., line 352 change "monitored" to obtained or recorded. Line 355 correct phrase "Tukey’s test a significance level of p-value". line 366 correct "highest" with higher. line 370-371 correct the syntax of phrase " the several biotransformation that involves the polyphenols".

Author Response

Response to reviewer 2

Manuscript ID: molecules-1157736

Type of manuscript: Article
Title: Chemical Composition, in Vitro Bioaccessibility and Antioxidant Activity of Polyphenolic Compounds from Nutraceutical Fennel Waste Extract

Point 1: Table 1 provide the full name for LOQ and LOD abbreviations, as it is not mentioned anywhere in text.

Response 2:  As suggested by reviewer 2, the authors added the missing information in the manuscript.

Point 2: lines340-341: correct the phrase"a previously work et al., [35] with some modifications". 

Response 2:  As suggested by reviewer 2, the authors changed the word “previously” to “previous”.

Point 3: for AR and NAR capsules suitable reference should be incorporated in text, preferably within paragraph 4.4, justifying the acid and non acid-resilient character of each capsule.

Response 3:  As suggested by reviewer 2, the authors added the missing information in the manuscript  as “The AR capsules in HPMC have been recognized as highly efficient in protecting components during gastric passage, while gelatin capsules are not resistant to the acidic gastric conditions [37].”

Point 4: minor corrections: line 28 correct " acid-gastric condition" to acidic gastric conditions

Response 4:  As suggested by reviewer 2, the authors changed “acid-gastric condition” to “acidic gastric conditions”

Point 5: Line 348 change 0.125 milliliters to 125 μl.

Response 5:  As suggested by reviewer 2, the authors changed “0.125 milliliters” to “125 μl”

Point 6: line 352 change "monitored" to obtained or recorded

Response 6:  As suggested by reviewer 2, the authors changed the word “monitored” to “recorded”.

Point 7: Line 355 correct phrase "Tukey’s test a significance level of p-value".

Response 7:  As suggested by reviewer 2, the authors changed the sentence “Tukey’s test a significance level of p-value≤0.05 was carried out to evaluate the differences among groups” to “The differences between mean values were evaluated by using Tukey’s test at the level of significance p-value≤0.05”

Point 8: line 366 correct "highest" with higher.

Response 8: As suggested by reviewer 2, the authors changed the word "highest" to “higher”

Point 9: line 370-371 correct the syntax of phrase " the several biotransformation that involves the polyphenols".

Response 9: As suggested by reviewer 2, the authors changed the sentence “Nevertheless, further studies useful to identify the several biotransformation that involves the polyphenols during in vivo gastrointestinal process are needed in order to better understand the possible benefits on human health” to “Nevertheless, further studies useful to identify the several metabolites generated during the gastrointestinal process are needed in order to better understand the possible benefits on human health.”

Point 10: Discussion section: Authors should include some references on in vitro digestion of hard capsules containing phenolics, and/or comparative studies with gelatin and hypromellose capsules, as a blank sample (non-encapsulated fennel powder) is not included in their study. Reference [15] (Izzo et al., 2020) could be the basis for that as well as Amrani-Allalou et al. (2021) etc.

Response 10: As suggested by reviewer 2, the authors added the missing information as “Interestingly, Izzo et al., [38] have been reported that red cabbage extract encapsulated in AR capsule after in vitro gastrointestinal digestion showed higher colon bioaccessibility when compared to the extract digested without a capsule Similar results have been observed by Amrani-Allalou et al., [39] who reported a strong decrease in TPC value during gastrointestinal digestion in non-encapsulated medicinal plant extracts compared to the same extracts containing in AR capsules.”.

Point 11: Lines 218-221: “As expected…digestion stages”. Improve the syntax of the sentence to facilitate readers’ comprehension.

Response 11: As suggested by reviewer 2, the authors changed the sentence “As expected, both oral and gastric bioaccessibility reported in this study from fennel extract contained in AR capsules was not relevant, suggesting that the capsules were not decomposed during these digestion stages. “ to “As expected, the oral and gastric bioaccessibility reported from AR samples was 0 mg GAE/g, suggesting that the capsules were not affected during these digestion stages.”

The authors thank reviewer 2 for evaluating our manuscript.

Reviewer 3 Report

Dear Authors,

After the review process, I have several comments: the authors should insert numerical data in the abstract; the authors should insert more new data about the bioactive potential of functional products  (e.g., compounds) and bioavailability of phenolic compounds in the introduction; the authors should insert in Page 2, second paragraph alternative methods to determine bioavailability and bioactivities of polyphenols after in vitro digestion (e.g., in vitro simulations), these data represents an essential part of the article; the article should insert references in all Materials and Methods sections; the authors should present a correlation between total phenolic content and in vitro antioxidant activity; the authors should comment a possible correlation between their data and in vivo data that valorized other food wastes; the authors should insert a control in the case of bioactivities.

Best regards!

Author Response

Response to reviewer 3

Manuscript ID: molecules-1157736

Type of manuscript: Article
Title: Chemical Composition, in Vitro Bioaccessibility and Antioxidant Activity of Polyphenolic Compounds from Nutraceutical Fennel Waste Extract

After the review process, I have several comments:

Point 1: The authors should insert numerical data in the abstract;

Response 1: As suggested by reviewer 3, the authors added the missing information in the abstract

Point 2: The authors should insert more new data about the bioactive potential of functional products  (e.g., compounds) and bioavailability of phenolic compounds in the introduction;

Response 2: As suggested by reviewer 3, the authors added the missing information in the introduction section as “ Scientific evidence suggests that the above-mentioned active compounds, in addition to their well-established antioxidant capacity, exert a wide range of other important biological properties such as anti-inflammatory and anticancer activities, as well as the ability to modulate cell signaling and gene expression in several experimental models [13], becoming good candidates for nutraceutical applications [14]. To fully exert their biological properties, polyphenols need to be available for absorption in the target tissue [15]. Moreover, several studies have shown that the absorption of polyphenols occurs mainly in the colon stage after metabolized by gut microbiota [16].”

Point 3: The authors should insert in Page 2, second paragraph alternative methods to determine bioavailability and bioactivities of polyphenols after in vitro digestion (e.g., in vitro simulations), these data represents an essential part of the article;

Response 3: As suggested by reviewer 3, the authors added the missing information in the discussion section as “The protocol used to simulate human gastrointestinal digestion in the present study was recently developed in the COST action INFOGEST network [33]. The above protocol is recognized as the most suitable method for comparing results across research teams. Since the fermentation of the large intestine is not covered in the INFOGEST protocol, the combined action of Viscozyme L and Pronase E has been used to simulate the colon digestion process [34]. Viscozyme L contains a mix of several carbohydrases such as xylanase, β-glucanase, hemicellulose, glucanase, arabinose, and cellulase, whereas Pronase E is a mixture of bacterial protease. Previous studies have proposed the use of Pronase E and Viscozyme L to reproduce the activity of the gut microbiota as an effective alternative to use of fecal inoculum [35]. On the other hand, several strategies have been reported to evaluate the polyphenol bioaccessibility after simulated gastrointestinal digestion. The use of immortalized cell lines such as Caco-2 represents the most common technique for assessing polyphenol bioaccessibility [36]. However, in order to obtain an overview of polyphenol bioaccessibility, in the present work TPC by the Folin-Ciocalteu assay was measured after all phases of gastrointestinal digestion. The results highlighted strong correlations among the data obtained from DPPH, ABTS, and FRAP assays and TPC values assessed during in vitro gastrointestinal digestion, supporting that the assayed tests provide some reliable information on the antioxidant compounds released by the nutraceutical formulations assayed during the simulated gastrointestinal process.”

Point 4: The article should insert references in all Materials and Methods sections;

Response 4: As suggested by reviewer 3, the authors added references in all Materials and Methods sections

Point 5: The authors should present a correlation between total phenolic content and in vitro antioxidant activity;

Response 5: As suggested by reviewer 3, the authors added data about correlation between total phenolic content and in vitro antioxidant activity

Point 6: The authors should comment a possible correlation between their data and in vivo data that valorized other food wastes;

Response 6: As suggested by reviewer 3, the authors added the missing information in the discussion section as “Nowadays, the valorization of food waste and by-products from agriculture and other food sectors is growing rapidly due to the well-known opportunities to recover a wide range of high added-value compounds [24]. Data obtained from several in vitro tests highlighted that various food waste and by-products, including olive mill wastewater, olive leaves, winemaking waste, brewing spent grains, apple pomace, among others, could be used as a sustainable source of bioactive molecules for cosmeceutical and nutraceutical applications [25].”

Point 7: The authors should insert a control in the case of bioactivities.

Response 7: This work aimed to investigate the bioaccessibility of total phenolic compounds and antioxidant capacity during simulated gastrointestinal digestion of two nutraceutical formulations containing aqueous-based extracts from fennel waste, identifying the best choice for this purpose. It would have been interesting to add to the analysis a control sample, the authors thank Reviewer 3 for the valuable advice and will adopt it in the next scientific work that will address this topic again. Moreover, the authors decided to include some references on in vitro digestion of hard capsules containing phenolics and comparative studies with some types of capsules, as a blank sample (non-encapsulated fennel powder) is not included in the study. As “Interestingly, Izzo et al., [38]have reported that red cabbage extract encapsulated in AR capsule after in vitro gastrointestinal digestion showed higher colon bioaccessibility when compared to the extract digested without a capsule. Similar results have been observed by Amrani-Allalou et al., [39] who reported a strong decrease in TPC value during gastrointestinal digestion in non-encapsulated medicinal plant extracts compared to the same extracts containing in AR capsules. These outcomes show that the AR capsules are able to protect the polyphenol fraction from the gastric condition, protecting the chemical characteristics of the bioactive molecules. Therefore, AR capsules can  represent a valid strategy to move active molecules to the target tissues for exerting their nutraceutical potential”

The authors thank reviewer 3 for evaluating our manuscript.

Round 2

Reviewer 3 Report

Dear Authors,

In general, you responded to my comments, and inserted only known data in the introduction, like ”Caco-2 represents the most common ...”. 

Best regards!